# Change in the Gut Microbiome and Immunity by *Lacticaseibacillus rhamnosus* Probio-M9

Meng Zhang,[a,b,c] Yan Zheng,[a,b,c] Zheng Sun,[a,b,c] Chenxia Cao,[a,b,c] Wei Zhao,[a,b,c] Yangshuo Liu,[a,b,c] Wenyi Zhang,[a,b,c] Heping Zhang[a,b,c]

[a]Key Laboratory of Dairy Biotechnology and Engineering, Ministry of Education, Inner Mongolia Agricultural University, Hohhot, People's Republic of China
[b]Key Laboratory of Dairy Products Processing, Ministry of Agriculture and Rural Affairs, Inner Mongolia Agricultural University, Hohhot, People's Republic of China
[c]Inner Mongolia Key Laboratory of Dairy Biotechnology and Engineering, Inner Mongolia Agricultural University, Hohhot, People's Republic of China

Meng Zhang, Yan Zheng, and Zheng Sun contributed equally to this work. Author order was determined by type of their contributions.

**ABSTRACT** With the exploding growth of the global market for probiotics and the rapid awakening of public awareness to manage health by probiotic intervention, there is still an active debate about whether the consumption of probiotics is beneficial for non-patients, which is due to the lack of systematic analysis based on time series multiomics data sets. In this study, we recruited 100 adults from a college in China and performed a random case-control study by using a probiotic (*Lacticaseibacillus rhamnosus* Probio-M9) as an intervention for 6 weeks, aiming to achieve a comprehensive evaluation and under-standing of the beneficial effect of Probio-M9 consumption. By testing advanced blood immunity indicators, sequencing the gut microbiome, and profiling the gut metabolome at baseline and the end of the study, we found that although the probiotic intervention has a limited impact on the human immunity and the gut microbiome and metabolome, the associations between the immunity indicators and multiomics data were strength-ened, and further analysis of the gut microbiome's genetic variations revealed inhibited generation of single nucleotide variants (SNVs) by probiotic consumption. Taken together, our findings indicated an underestimated influence of the probiotic, not on altering the microbial composition but on strengthening the association between human immunity and commensal microbes and stabilizing the genetic variations of the gut microbiome.

**IMPORTANCE** Although the global market for probiotics is growing explosively, there is still an active debate about whether the consumption of probiotics is beneficial for nonpatients. In this study, we recruited 100 adults from a college in China and per-formed 6 weeks of intervention for half of the volunteers. By analyzing the time series multiomics data in this study, we found that the probiotic intervention (i) has a limited effect on human immunity or the global structure of the gut microbiome and metabo-lome, (ii) can largely influence the correlation of the development between multiomics data and immunity, which was not able to be discovered by conventional differential abundance analysis, and (iii) can inhibit the generation of SNVs in the gut microbiome instead of promoting it.

**KEYWORDS** probiotic, *Lactobacillus rhamnosus*, multiomics, single nucleotide variants

Probiotics, defined as live microorganisms that when administered in adequate amounts confer a health benefit on the host (1, 2), are one of the most clinically feasible approaches for regulating the gut microbiome and metabolome (3) and have great poten-tial to be therapeutic targets in many human diseases. For example, evidence from previ-ous studies has suggested that modulation of the intestinal microbiota and microbial metabolites by consuming probiotics is beneficial in relieving chronic diseases (4–6), improving being overweight and obesity (7–10), delaying nonalcoholic fatty liver disease (11), treating gastrointestinal diseases (12–17), regulating neurological diseases (18–20),

Address correspondence to Heping Zhang, hepingdd@vip.sina.com, or Wenyi Zhang, zhangwenyizi@163.com.

The authors declare no conflict of interest.

reducing subjective stress (21), improving immunity (22), preventing oxidative stress and inflammation (23), and preventing antibiotic-induced *Clostridium difficile* infection (24–26).

However, an active debate has been going on for years about whether the consumption of probiotics is beneficial for nonpatients in quality-of-life improvement and disease prevention (27–29). Although previous studies supported a beneficial effect of probiotics in enhancing immunity against the common cold, which can reduce the incidence (30), duration (31), and symptoms (32) of the common cold (28, 33, 34), some reported that the effects of the probiotics on the immune system and gastrointestinal symptoms in nonpatients are limited (29, 35). One approach by which probiotics impact the host immune system is through the microbial metabolism that arises from intestinal microbiota catabolism (36–39): e.g., functional metagenomic studies have identified the associations between host proinflammatory cytokines and microbial tryptophan and palmitoleic acid metabolic pathways (37, 38, 40–42). Moreover, previous studies have illustrated that the gut microbiota and metabolites activate the host immune system, thereby increasing the expression of endocrine peptides and promoting host metabolic homeostasis (37, 43, 44). In addition, characterization of stable and changeable genetic components in the gut microbiome is crucial for further understanding the role of the gut microbiome in human health and phenotypic changes (45), which has rarely been investigated in probiotic studies. Therefore, a comprehensive evaluation of the role of the gut microbiome, the change in microbial metabolites, and variations in their genetic structure in the beneficial effect of probiotic consumption is still urgently warranted due to the lack of systematic analysis of objective immunity indicators and time series multiomics data in previous studies.

To address the above questions, we recruited 100 adult volunteers in China and randomly assigned them into two groups. Then, *Lacticaseibacillus rhamnosus* Probio-M9 ($5.0 \times 10^{10}$ CFU per day) was administered as the probiotic intervention in one of the groups for 6 weeks. We sought to explore how *L. rhamnosus* Probio-M9 would impact the gut microbiome, gut metabolome, host immune response, and its variants in the genomic structure after consumption in humans. So, we performed advanced immune function tests and metagenomic sequencing and metabolomic profiling using the blood and stool samples collected at the beginning (baseline) and end of the trial (endline). By analyzing the time series multiomics data in this study, we first found that probiotic intake has a limited effect on human immunity and gut microbial composition but can bring about a convergence of the gut microbiome and metabolome. This interested us in further exploring the association between the multiomics data and human immunity, and we found altered associations between the multiomics data and immunity during the probiotic intervention. Finally, we sought to explain such alterations by analyzing the genomic variation of the gut microbiome (such as single nucleotide variants [SNVs]). Interestingly, inhibited SNVs in the gut microbiome were identified during the probiotic intervention. Our study underlined an underestimated influence of probiotic consumption, not on the gut microbial composition but on the association between human immunity and the gut microbiome, as well as the generation of SNVs. Specifically, we found that Probio-M9 has an outstanding performance in tightening the association between immunity and the gut microbiome and stabilizing the genetic makeup in the gut microbiome of nonpatients.

## RESULTS

**Participants.** To quantitatively measure the influence of probiotic consumption, we recruited 100 adult volunteers from Inner Mongolia Agricultural University in China (see Materials and Methods). The age and body mass index (BMI) of the participants ranged from 21 to 41 years old (mean, 26.13) and 18.51 to 29.98 (mean, 22.10). All participants were self-reported disease-free and validated by the immune function test with no inflammation symptoms. Moreover, all participants had no dietary restrictions and had eaten for more than 5 days at the same canteen on the campus. The volunteers were then randomly assigned to either the placebo group or probiotic group by

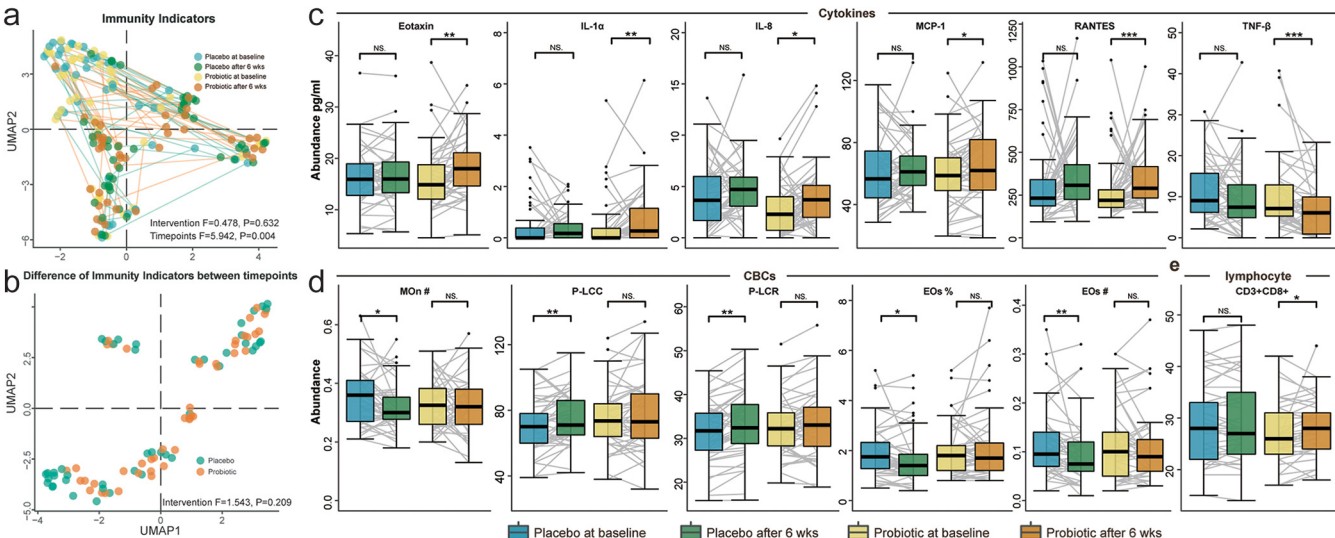

**FIG 1** Comparison of the advanced immune function test results. (a) The UMAP of the advanced immune test results is based on 69 immunity indicators. The difference between the probiotic group (light yellow and orange) and the placebo group (light blue and green) was not significant at the beginning or end of the intervention. (b) In the comparison (illustrated by the UMAP) of the changes of 69 immunity indicators during the intervention, no significant signal was detected. Green dots refer to the changes of indicators within individuals in the placebo group, while orange refers to the changes in the probiotic group. (c) Comparison of differential cytokines in the two groups at different time points; (d) comparison of differential CBCs in the two groups at different time points; (e) comparison of differential lymphocytes in the two groups at different time points. Significance levels: *, $P < 0.05$; **, $P < 0.01$; ***, $P < 0.001$; NS, not significant.

gender to ensure an equal number of females and males in each group (20 males and 30 females in the placebo group, 21 males and 29 females in the probiotic group). At the beginning of the intervention, there are no significant differences in age or BMI between the two groups (Student's $t$ test; $P = 0.133$ for age and $P = 0.068$ for BMI). By the end of the probiotic intervention, five individuals in each group reported antibiotic intake or noncompliance: thus, a total of 90 participants (45 participants in each group) were included in the final analysis. Finally, 180 whole-blood samples and 180 stool samples were collected at baseline and end of the intervention, and then whole-blood-based advanced immune function tests, which include complete blood count (CBC), lymphocyte, and circulating cytokines, see Table S1 in the supplemental material, whole-metagenome sequencing (WMS) and metabolome profiling (of stool) were performed (see Materials and Methods).

**The effect of probiotic consumption on human immunity.** First, in order to detect the presence of *Lacticaseibacillus rhamnosus* Probio-M9 in the feces, we added the reference genome of *Lacticaseibacillus rhamnosus* into the database of Kraken2 and performed profiling. Although the identification at the strain level is not well supported by Kraken2, we found that *Lacticaseibacillus rhamnosus* in the probiotic group is significantly higher than that in the placebo group (Fig. S1), suggesting the effectiveness of the administration of Probio-M9 for the volunteers in the treatment group. To explore the impact of probiotic consumption on human immunity, we first compared the advanced immune function test results between the two groups at the beginning and end of the intervention (Fig. 1a). By visualizing the comparison of 69 immunity indicators (Table S1) using UMAP (46), we found that there was no significant difference at baseline (permutational multivariate analysis of variance [PERMANOVA]; $P = 0.486$) or the end of the trial (PERMANOVA; $P = 0.858$) between the two groups. Moreover, neither of the comparisons of the changes in immune function test results during the intervention was found to be significant between the two groups (PERMANOVA; $P = 0.209$) (Fig. 1b), which suggests that the probiotic has a limited impact on overall human immunity through 6 weeks of intervention.

We then sought to investigate if any of the 69 indicators of immunity had been altered during the trial by performing differential abundance analysis: e.g., we compared

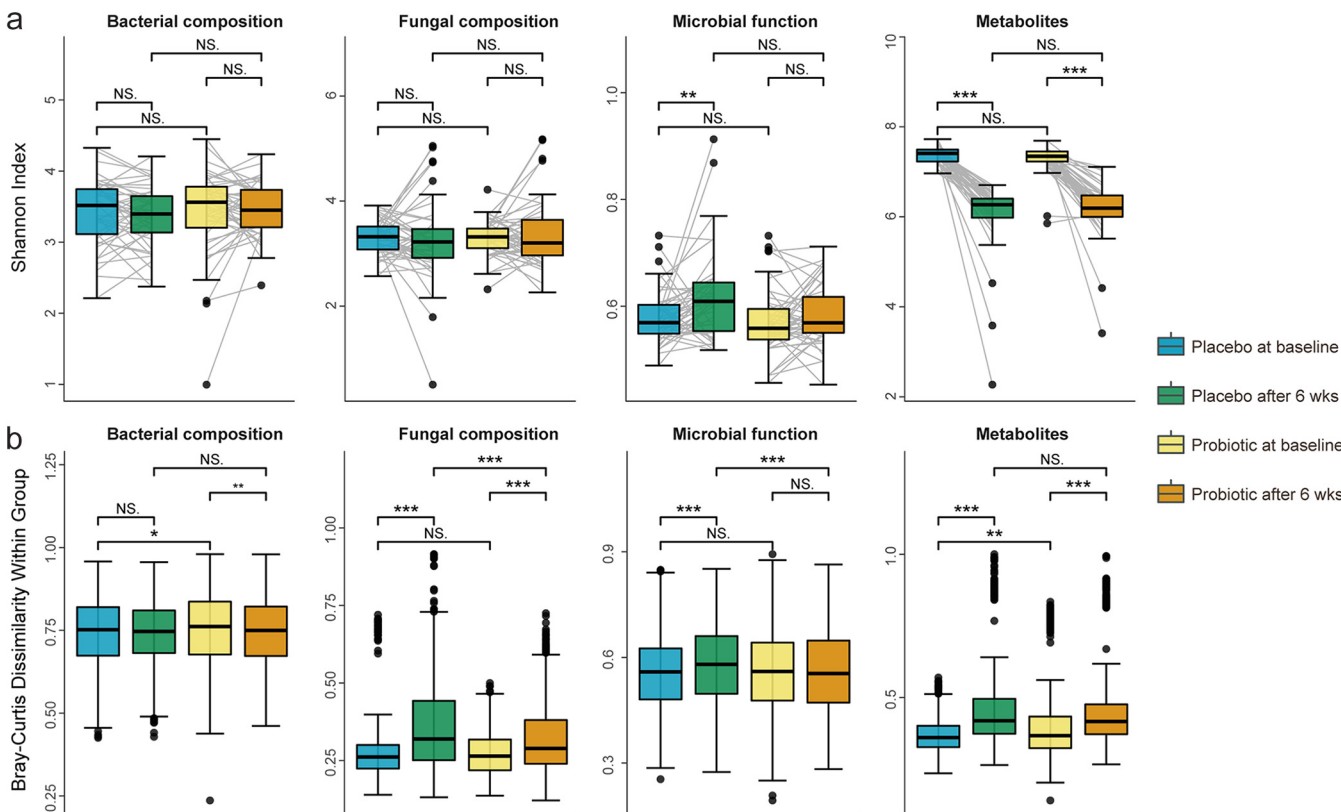

**FIG 2** Comparison of $\alpha$ diversity and within-group BC dissimilarity. (a) Comparison of $\alpha$ diversity between the two groups at different time points. $\alpha$ diversity is measured by the Shannon index based on the bacterial composition, fungal composition, microbial function, and metabolites separately. (b) Comparison of convergence (within-group BC dissimilarity) between the two groups at different time points. Blue and green boxes refer to the placebo group at baseline and the end of the intervention, while yellow and orange boxes refer to the probiotic group at baseline and the end of the intervention. Significance levels: *, $P < 0.05$; **, $P < 0.01$; ***, $P < 0.001$; NS, not significant.

the immunity indicators between the two time points from the same individual and then filtered out the indicators that changed along with the time (see Materials and Methods). In cytokines, compared with the placebo group, eotaxin, interleukin-1$\alpha$ (IL-1$\alpha$), IL-8, monocyte chemoattractant protein 1 (MCP-1), and RANTES were found to be increased by 15.48% ($P = 0.001$), 39.61% ($P = 0.006$), 31.06% ($P = 0.013$), 8.94% ($P = 0.013$), and 23.95% ($P = 0.014$), while tumor necrosis factor beta (TNF-$\beta$) was found to be decreased by 38.58% ($P = 0.001$) after the probiotic intervention (Fig. 1c). However, in CBCs, we did not identify any significant changes in the probiotic group. Instead, the number of eosinophils, percentage of eosinophils, and number of monocytes were found to be decreased by 18.32% ($P = 0.008$), 11.86% ($P = 0.015$), and 5.77% ($P = 0.041$), and the platelet large cell count (P-LCC) and platelet large cell ratio (P-LCR) were found to be increased by 5.67% ($P = 0.006$) and 5.57% ($P = 0.003$) in the placebo group (Fig. 1d). As for lymphocytes, we identified only one significant difference from CD3[+] CD8[+] (increased by 4.45%; $P = 0.016$) in the probiotic group (Fig. 1e). In summary, although 12 out of 69 indicators of immunity were found to be altered during the probiotic intervention, the magnitude of the changes in those indicators revealed a slight alteration of immunity.

**Unaffected microbial diversity of the gut microbiome and metabolome.** In previous microbiome studies, $\alpha$ diversity was associated with a healthier gut microbiome (47–49) and altered $\beta$ diversity in the gut microbiome was discovered in many diseases (48–50). This interested us in comparing the $\alpha$ and $\beta$ diversities of the gut microbiome and metabolome at baseline and at the end of the probiotic intervention.

For $\alpha$ diversity (Fig. 2a), the Shannon index of bacterial composition (by *de novo* assembly and binning [see Materials and Methods]), fungal composition (by Kraken2

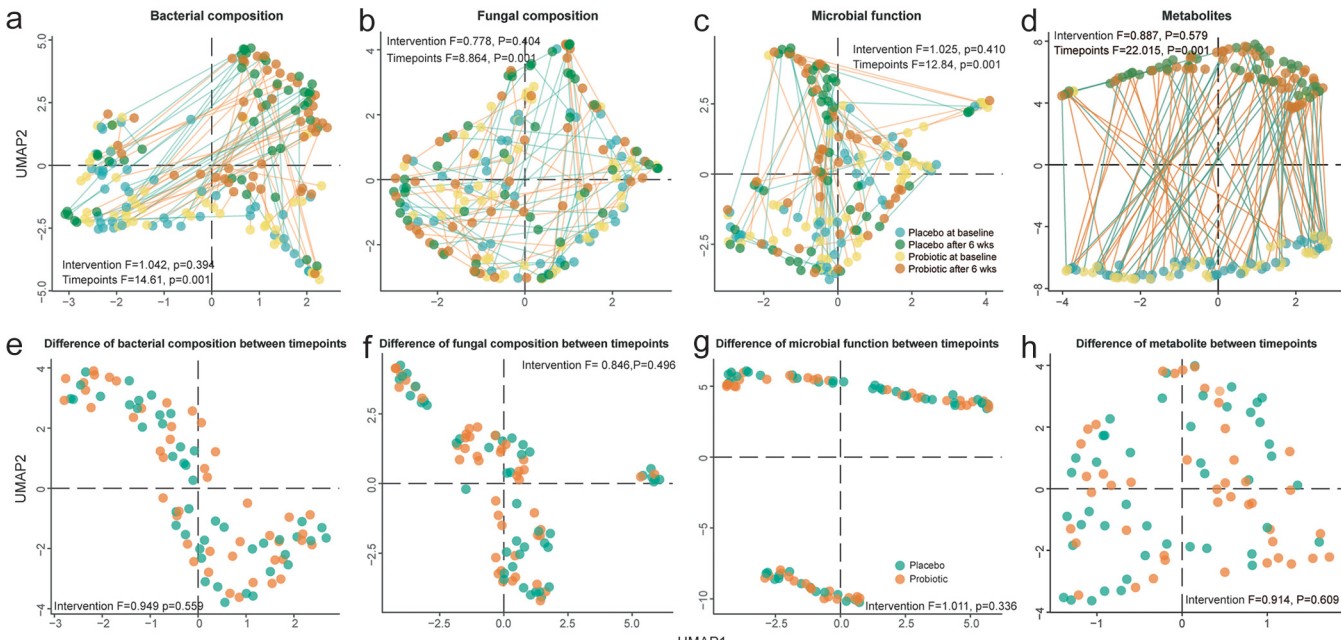

**FIG 3** β diversity during the intervention. The UMAP of multiomics data is based on BC dissimilarity of (a) bacterial composition, (b) fungal composition, (c) microbial function, and (d) metabolites separately. The difference between time points or groups (intervention) was determined by PERMANOVA. Moreover, the rates of development of the (e) bacterial composition, (f) fungal composition, (g) microbial function, and (h) metabolites during the trial were also compared between the two groups and tested by PERMANOVA.

[Materials and Methods]), and microbial function (by HUMAnN3 [Materials and Methods]) remained unchanged in the probiotic group (baseline versus endline; $P$ = 0.640, 0.746, and 0.401, respectively) after 6 weeks of intervention. (The profiling results at the species level can be found in Table S2). Although the Shannon index of metabolites (by Progenesis QI2 [Materials and Methods]) decreased in both groups during the intervention (baseline versus endline; $P$ = 5.684e−14 and 5.684e−13 for the placebo group and probiotic group, respectively), no significant difference was found between the two groups at the beginning or end of the trial ($P$ = 0.385, 0.542, 0.086, and 0.816 for bacterial composition, fungal composition, microbial function, and metabolites, respectively).

As for the β diversity, we visualized the comparison by UMAP (46) and found that regardless of bacterial composition (Fig. 3a), fungal composition (Fig. 3b), microbial function (Fig. 3c), or metabolites (Fig. 3d), there was no significant difference between the two groups at baseline (PERMANOVA; $P$ = 0.792, 0.624, 0.501, and 0.489, respectively) or at the end of the trial ($P$ = 0.924, 0.868, 0.332, and 0.393, respectively). Instead, we found the time point has a much larger effect size than the probiotic intervention on the gut microbiome and metabolome: e.g., PERMANOVA $F$ values for time point are 14.61, 8.864, 12.84, and 22.02 ($P$ = 0.001), respectively, for bacterial composition, fungal composition, microbial function, and metabolites, while for probiotic treatment, the $F$ values are 1.042, 0.778, 1.025, and 0.887 ($P$ > 0.05), respectively. Further evaluation of the influence of the probiotic intervention on the development of the gut microbiome and metabolome also illustrated a nonsignificant change (Fig. 3e to h) when comparing the development (difference) of multiomics data between two time points by different groups (PERMANOVA; $P$ = 0.559, 0.496, 0.336, and 0.609 for bacterial composition, fungal composition, microbial function, and metabolites, respectively). All of the above findings indicate that the probiotic had a limited impact on microbial diversity of the gut microbiome and metabolome through 6 weeks of intervention.

**Tightened convergence of the gut microbiome and metabolome.** We then sought to investigate whether the convergence (based on the Bray-Curtis [BC] dissimilarity) was affected by probiotic consumption (Fig. 2b). By comparing within-group BC

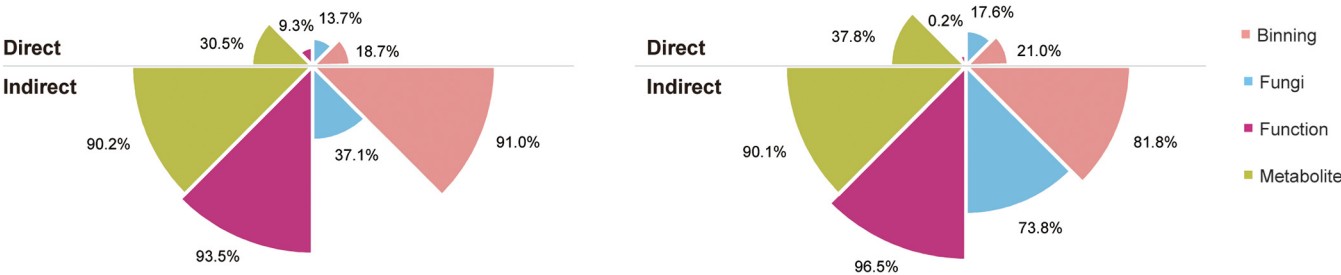

**FIG 4** Comparison of the direct influence and indirect influence of the probiotic intervention on the gut microbiome and metabolome. The upper rose charts reflect the direct influence (e.g., the differentially abundant taxa, functions, or metabolites induced by the probiotic intervention), while the lower rose charts represent the indirect influence (e.g., the taxa, functions, or metabolites whose development is significantly correlated with the change of the immunity during the probiotic intervention). The left panel illustrates the ratio between the number of the impacted features and the number of all identified features (from bacterial composition, fungal composition, microbial function, and metabolites), while the right panel is the proportion of the impacted features.

dissimilarity between the probiotic group and placebo group, we found the following. (i) For bacterial composition, within-group BC dissimilarities were found to be decreased in the probiotic group (Wilcoxon signed-rank test; $P = 0.003$), while nonsignificant change was detected in the placebo group ($P = 0.570$). (ii) For fungal composition, although within-group BC dissimilarities were both increased in the two groups, the probiotic group has significantly lower within-group BC dissimilarity than the placebo group at the end of the intervention (Wilcoxon rank sum test; $P < 0.001$). (iii) For microbial function, we found a similar within-group BC dissimilarity between the two groups at baseline ($P = 0.320$) but a significantly lower within-group BC dissimilarity in the probiotic group than the placebo group at the end of the intervention ($P = 2.0e-04$). (iv) For metabolites, although within-group BC dissimilarities were increased in both of the two groups, the probiotic group has a significantly higher within-group BC dissimilarity at baseline ($P = 0.003$) while such significance was not able be observed at the end of the study. This suggests that the intervention of probiotic tightened the convergence of the gut microbiome and gut metabolome.

**The underestimated influence of probiotic intake on the association between multiomics data and immunity.** The convergence of the gut microbiome and metabolome raised the question of whether the change in taxa and metabolites is related to the change of immunity during the probiotic intervention. To address this question, we first correlated individual's changes in multiomics data with the changes in immune function test results during the study and then compared the significant immunity feature (immunity indicators with taxa, functions, or metabolites) correlations (defined by a Spearman coefficient of $<0.05$) between the two groups and removed the same correlations identified in both groups (Materials and Methods). We found that the changes of 91.0% (81.8%) in bacterial composition, 37.1% (73.8%) in fungal composition, 93.5% (96.5%) in microbial function, and 90.2% (90.1%) in metabolite number (proportion) of taxa, functions, or metabolites were associated with the changes in immune function test results (Fig. 4; indirect influence), suggesting a large influence of the probiotic intervention on the association of development between the gut microbiome (and metabolome) and immunity. Moreover, analysis of the association between the gut microbiome and immunity revealed a higher proportion of bacterial (Wilcoxon rank sum test; $P = 1.42e-07$) and fungal ($P = 0.046$) composition correlated with immune test results at the end of the trial compared to the baseline (data no shown).

Conventionally, differential abundance analysis is employed to evaluate the direct influence and consequence of the probiotic intervention. However, we found that such direct influence caused by the probiotic intervention is much less than probiotic consumption's indirect influence on the association between the gut microbiome and immunity. For example, we found that the percentagesr (and proportion) of differentially abundant taxa, functions, and metabolites are only 18.7% (21.0%; bacterial composition), 13.7% (17.6%; fungal composition), 9.3% (0.2%; microbial function), and 30.5%

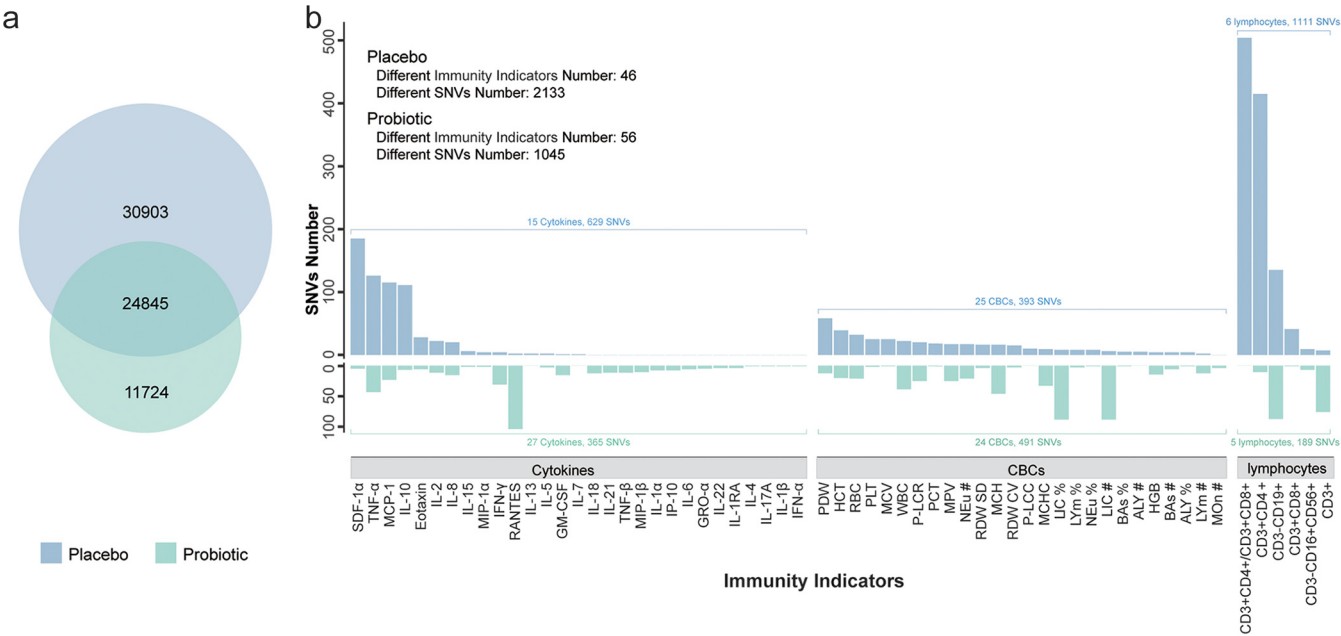

**FIG 5** Comparison of the SNVs generated during the intervention in the probiotic group and placebo group. (a) The Venn diagram of SNVs was generated in the placebo group and probiotic group. The intersection is considered to be the consequence of time development instead of different treatment. (b) Comparison of immunity-related SNVs between the two groups. The unique SNVs in the probiotic or placebo group were associated with the changes of 69 immunity indicators and are illustrated by bar plots.

(37.8%; metabolites) between the two groups (Fig. 4; direct influence), which is much smaller than the indirect influence (Fig. 4; indirect influence). This indicates that the influence of the probiotic intervention on the gut microbiome and metabolome can be highly underestimated because the indirect influence on enhancing the association between the multiomics data and immunity is beyond the scope of conventional differential abundance analysis.

**Probiotic intake inhibits the generation of SNVs in the gut microbiome.** To further understand the mechanism of altered association between the multiomics data and immunity, we investigated whether such indirect impact on the gut microbiome is raised through altering the genetic structure: e.g., the SNVs in the microbial genomes. First, Bowtie2 and SAMtools were employed to identify candidate SNVs from species-level genome bins (SGBs) for each sample (with a minimum quality of 60 using VCFtools and BCFtools [Materials and Methods]). Second, the candidate SNVs from the same individual at different time points were compared to further screen the time- and treatment-related candidate SNVs. Third, we compared the time- and treatment-related candidate SNVs between the two groups to select only treatment-related candidate SNVs by filtering out the SNVs related to time (Fig. 5a) (Materials and Methods). Interestingly, we found 11,724 unique SNVs in the probiotic group, while the placebo group contains 30,903 unique SNVs (Table S3), suggesting that probiotic consumption did not accelerate but indeed inhibited the generation of SNVs compared to placebo consumption.

We then correlated the unique SNVs with the changes in immune function test results to further explore the roles of the probiotic-induced SNVs in the association between the gut microbiome and immunity (Materials and Methods). We found that although a higher number of unique SNVs was associated with the changes in immune test results in the placebo group than in the probiotic group (2,133 versus 1,045), the number of the related immune indicators was higher in the probiotic group (46 versus 56) (Fig. 5b). Specifically, for the SNVs related to the change of cytokines, only 15 cytokines (with SDF-$\alpha$, TNF-$\alpha$, MCP-1, and IL-10 as the most-associated items) were found to be correlated with 629 SNVs in the placebo group, while 365 SNVs were found to be correlated with 27 cytokines (with RANTES as the most-associated item) in the probiotic group. As for the SNVs correlated with CBCs in the placebo group and probiotic

**TABLE 1** The top 10 SNVs abundant species in the probiotic and placebo group

| Annotation of SGBs by GTDB | SNV no. | | | SNV rank | | | Mean % relative abundance | |
|---|---|---|---|---|---|---|---|---|
| | Probiotic | Placebo | Sum | Probiotic | Placebo | Sum | Probiotic | Placebo |
| *Agathobacter faecis* | 2,022 | 60 | 2,082 | 1 | | 10 | 1.18 | 3.43 |
| *Clostridium* sp003024715 | 1,818 | 429 | 2,247 | 2 | | 7 | 0.44 | 0.61 |
| *Gemmiger qucibialis* | 1,432 | 878 | 2,310 | 3 | | 6 | 0.61 | 0.47 |
| *Bifidobacterium adolescentis* | 460 | 343 | 803 | 8 | | 14 | 1.92 | 1.67 |
| ER4 sp000765235 | 343 | 213 | 556 | 10 | | 15 | 0.37 | 0.23 |
| *Roseburia* sp900552665 | 1,290 | 1,507 | 2,797 | 4 | 8 | 5 | 0.98 | 1.06 |
| *Fusicatenibacter saccharivorans* | 746 | 992 | 1,738 | 5 | 10 | 11 | 2.67 | 2.75 |
| *Anaerobutyricum hallii* | 548 | 1,635 | 2,183 | 6 | 6 | 8 | 1.39 | 1.92 |
| *Blautia_A wexlerae* | 472 | 1,029 | 1,501 | 7 | 9 | 13 | 2.81 | 3.23 |
| *Blautia_A massiliensis* | 351 | 3,766 | 4,117 | 9 | 3 | 2 | 1.72 | 1.63 |
| *Faecalibacterium prausnitzii_G* | 112 | 4,385 | 4,497 | | 1 | 1 | 1.26 | 1.12 |
| *Roseburia intestinalis* | 9 | 4,045 | 4,054 | | 2 | 3 | 0.86 | 0.69 |
| *Anaerostipes hadrus* | 79 | 2,781 | 2,860 | | 4 | 4 | 1.68 | 1.98 |
| *Phascolarctobacterium faecium* | 1 | 2,090 | 2,091 | | 5 | 9 | 0.18 | 0.29 |
| *Bifidobacterium pseudocatenulatum* | 18 | 1,561 | 1,579 | | 7 | 12 | 2.19 | 3.36 |

group (25 and 24 CBCs, respectively), we also found differential association patterns: e.g., the most-associated item is platelet distribution width (PDW) in the placebo group, while the percentage of large immature cells (LICs) and number of large LICs are the most-related items in the probiotic group. Moreover, we identified more SNVs (491) in the probiotic group than in the placebo group (393) that were related to the CBCs. Interestingly, between the two groups, the most significant difference in the association between immune function test results and SNVs comes from lymphocytes: e.g., 1,111 SNVs in the placebo group were found to be correlated with six lymphocytes (the most-associated items are $CD3^+$ $CD8^+$ and $CD3^+$ $CD4^+$), while there are only 189 SNVs correlated with five lymphocytes (the most-associated items are $CD3^-$ $CD19^+$ and $CD3^+$) in the probiotic group.

To better understand the generation of those unique SNVs at the level of taxonomic structure, we then categorized the SNVs by their original species and compared the top 10 SNV abundant species between the two groups (Table 1). In the probiotic group, we found that *Agathobacter faecis*, *Clostridium_Q* sp003024715, and *Gemmiger qucibialis* have the most dramatic change in SNVs compared to those in the placebo group (2,022, 1,818, and 1,432 versus 60, 429, and 878, respectively). In the placebo group, *Faecalibacterium prausnitzii_G*, *Roseburia intestinalis*, *Anaerostipes hadrus*, *Phascolarctobacterium faecium*, and *Bifidobacterium pseudocatenulatum* were found to have the most significant change (4,385, 4,045, 2,781, 2,090, and 1,561 versus 112, 9, 79, 1, 18 in the placebo group and probiotic groups, respectively). Moreover, an overlap was found in the top 10 SNV abundant species between the two groups, such as *Roseburia* sp900552665 (1,290 and 1,507 in the probiotic and placebo groups), *Fusicatenibacter saccharivorans* (746 and 992), *Anaerobutyricum hallii* (548 and 1,635), *Blautia_A wexlerae* (472 and 1,029), and *Blautia_A massiliensis* (351 and 3,766), which contain relatively high SNVs in both groups. Notably, the unique SNVs in the placebo group were supposed to be generated without probiotic consumption but were inhibited in the probiotic group, while the unique SNVs in the probiotic group were induced by the probiotic intervention, which was not able to be detected by taking a placebo. All of the above findings indicate that inhibition is a major effect of the probiotic intervention on the SNVs in the gut microbiome, and *Faecalibacterium prausnitzii_G*, *Blautia_A massiliensis*, and *Roseburia intestinalis* are the most sensitive species for the probiotic intervention.

## DISCUSSION

The size of the global probiotics market was valued at $58.17 billion (U.S. dollars [USD]) in 2021 and is expected to expand at a compound annual growth rate of 7.50%

from 2021 to 2030 (51). However, the benefits and feasibility of probiotic consumption in nonpatients remain uncertain (27–29, 35) due to the vast differences between the microbiomes of individuals (which arise through a complex combination of environmental, genetic, and lifestyle factors) (27) and the lack of an internationally accepted consensus definition of a normal or a healthy fecal microbial community (52). In addition, stability is critical in understanding the role of a dysbiosis microbiome in human diseases, and notably, microbial genetic variability provides an extra layer of information that is independent of microbial abundance (45). However, in previous studies (i) subjective self-report scales and questionnaires have been primarily used as measurements, (ii) advanced blood immune test and longitudinal multiomics data have seldom been applied together to provide comprehensive evaluation, and (iii) microbial genetic variations have rarely been evaluated. Therefore, there is an urgent need for understanding the benefits of probiotics on nonpatients through immunity indicators, the gut microbiome, and microbial metabolism and using time series data in a quantitative and systematic way.

The functional immune system is a central determinant of survival of the host organismal health from environmental pathogens, viruses, or chemicals (53, 54). Although there was just slightly altered overall human immunity after the probiotic intervention compared to the placebo, six cytokines, eotaxin, IL-1$\alpha$, IL-8, MCP-1, RANTES, and TNF-$\beta$, and one type of lymphocyte, CD3$^+$ CD8$^+$ T cells, showed significant improvements within the normal range, which is in line with previous findings that the T lymphocytes and cytokine interleukin are activated after probiotic consumption (28, 31, 55, 56). Complex ecological communities are generally thought to be more stable and resilient, and gut microbial diversity is often used as a proxy for human health (57–59). Even though 6 weeks of *Lacticaseibacillus rhamnosus* Probio-M9 consumption did not dramatically affect the overall gut microbiome and metabolites, it reduced the population heterogeneity of individuals in view of the multiomics perspective. To better understand such impacts on the host, we innovatively evaluated the probiotic's effects by categorizing them as direct effects and indirect effects. As expected, we found a significantly underestimated indirect effect of probiotics on the other intestinal microorganisms, metabolites, and immune indicators, underling the intermediary effect of Probio-M9 that regulates intestinal microbes and microbial metabolites and eventually affect the host immune system.

In this study, we found that the probiotic intervention (i) has a limited effect on human immunity or the global structure of the gut microbiome and metabolome, which agrees with previous studies (27–29, 35), (ii) can largely influence the correlation of the development between multiomics data and immunity, which was not able to be discovered by conventional differential abundance analysis, and (iii) inhibits the generation of SNVs in the gut microbiome instead of promoting it.

To our knowledge, the present research is the first to report the underestimated alteration of association between immunity and the gut microbiome/metabolome, as well as the suppressed SNVs by probiotic consumption in nonpatients. Despite many studies that have been conducted on *Lacticaseibacillus rhamnosus* Probio-M9 (such as its isolation process, biochemical characteristics [60], and mouse [61–63], and small-scale human experiments [64] examining its beneficial effects on hosts), the comprehensive evaluation of Probio-M9 using multiomics data collected from different time points, as presented in this article, greatly compensates for the gaps in probiotic research and significantly increases our understanding of the mechanisms behind the probiotic beneficial effects. However, there are some limitations to this study. For example, metaproteomic and metatranscriptomic data were not included in the evaluation, which may lead to a missing evaluation and understanding of transcription and translation processes induced by the probiotic intervention. On the other hand, blood and stool samples were not included in the middle of the intervention and after the washing-out period, which may provide details about the trajectory and resistance of the probiotic colonization. In summary, by demonstrating that Probio-M9 can tighten

the association between immunity and the gut microbiome and stabilize the genetic variations in the gut microbiome, we revealed a heavily underestimated influence of probiotic consumption, which provides us with a new perspective for comprehensively understanding and evaluating the efficacy of probiotics.

## MATERIALS AND METHODS

**Experimental design.** Between December 2018 and January 2019, 100 self-reported disease-free individuals (41 males and 59 females) were recruited for a randomized, double-blind, case-control probiotic intervention at Inner Mongolia Agricultural University in China. All participants were randomly assigned to either a placebo or probiotic group by gender to ensure roughly equal amounts of males and females were in each group. By the end of the probiotic intervention, a total of 90 participants (45 participants in each group) were included in the final analysis. Although no food restrictions were placed during the trial, suggestions on total calorie intake, carbohydrate, protein, fat, vitamins, and minerals (according to Chinese dietary guidelines from 2016 [65]) were given to the participants to minimize the confounding factors from diet.

The exclusion criteria for subjects included (i) taking long-term medications due to certain severe illnesses (e.g., organic or systemic diseases), (ii) suffering from chronic respiratory allergy (defined as taking allergy medication daily), (iii) suffering from severe recurrent gastrointestinal disease, (iv) undergoing pregnancy or lactation, (v) catching a cold on the first day of the intervention, (vi) suffering from immunosuppressive disease or having had immunosuppressive treatment in the past year, (vii) having received antibiotic treatment within 2 months before the intervention start, (viii) having an allergy to any ingredients contained in the probiotic product (Yishiyou produced by Scitop Bio), (ix) taking corticosteroids currently, or (x) taking any probiotic products regularly 3 weeks before starting the intervention. No participants reported any suboptimal health status, such as symptoms of anxiety, stress, heartburn, indigestion, nausea, vomiting, diarrhea, and constipation at the end of the trial.

**Probiotic usage and sample collection.** *Lacticaseibacillus rhamnosus* Probio-M9 ($5.0 \times 10^{10}$ CFU per day) was used for the probiotic intervention; Probio-M9 is a probiotic isolated from healthy women's breast milk (60, 66, 67). We selected this strain because it has no transferable antibiotic resistance genes (60) and is potentially beneficial for easing the symptoms of anxiety, stress, and gastrointestinal distress (64, 68). However, little effort has been made to comprehensively understand its mechanism: e.g., its changes in the genomic structure after administration. In addition, *Lacticaseibacillus rhamnosus* Probio-M9 is commercially available (the brand Yishiyou produced by Scitop Bio was used, which contained pure *Lacticaseibacillus rhamnosus* Probio-M9) and was inspected by the quality inspection department of China for its safety, concentration, and purity (2018061105). On the other hand, the placebo contained only maltodextrin and strawberry powder. All sachets were identical in taste and appearance. All of the volunteers in the probiotic group consumed a 2-g package containing *Lacticaseibacillus rhamnosus* Probio-M9 daily (after breakfast) for 6 weeks, while the volunteers in the placebo group took the placebo of the same weight once daily for 6 weeks.

Whole-blood and stool samples were collected at baseline and end of the intervention from all 90 participants. Specifically, whole-blood samples were collected by doctors for advanced immune function tests (e.g., complete blood count and lymphocyte). CBCs and lymphocytes were tested immediately after the blood samples were collected, while the remaining blood samples were stored at −80°C for assays of circulating cytokines in the Inner Mongolia Agricultural University. Stool samples were collected by the volunteers themselves following the instructions described previously (68, 69). All stool samples were transferred into −80°C refrigerators the day after collection before being sent out for metagenomic sequencing and metabolomic profiling in the Inner Mongolia Agricultural University.

**Advanced blood immune function test.** The concentrations of plasma inflammatory cytokines were analyzed using the Human Cytokine & Chemokine 34-plex Procarta PlexPanel (Thermo Fisher Scientific, USA) with 25 $\mu$L plasma from each blood sample. The tests were performed according to the manufacturer's instructions, and the results were reported by the Luminex 200 multiplexing instrument (Bio-Rad, USA). A total of 34 cytokines were analyzed. In addition, 29 CBC indicators and six lymphocytes were reported in the Affiliated Hospital of Inner Mongolia Agricultural University. For details of all 69 immunity-related indicators, please refer to Table S1 in the supplemental material.

**Metagenomic sequencing and data processing.** For metagenomic sequencing, the Illumina HiSeq 2500 platform was employed as previously described (69). In brief, DNA extraction was performed using 4.5-mL stool suspensions that had been thawed on ice for 1 h after vortexing. A standard stool DNA extraction was conducted using the Qiagen DNA stool minikit (Qiagen) following the manufacturer's instructions. Genomic DNA quality and concentration were analyzed by gel electrophoresis and with a NanoDrop 8000 spectrophotometer (Thermo Electron Corp., Waltham, MA, USA), respectively. The final DNA concentration was above 100 ng/$\mu$L, and the 260-nm/280-nm ratio was between 1.8 and 2.0. Libraries were prepared with a fragment length of ~300 bp. Paired-end reads were generated using 150 bp in the forward and reverse directions. After removal of the low-quality and human-derived DNA by Kneaddata (v.0.10.0) (70), the metagenomic sequencing data remain at a mean of 25,191,034 high-quality paired-end reads in each sample.

The bacterial composition was profiled by a *de novo* assembly and binning method following Saheb Kashaf's protocol (71). Briefly, we (i) assembled metagenomic short clean reads into contigs using MEGAHIT (v.1.2.2) (72), (ii) selected metagenome-assembled genomes (MAGs) using MaxBin 2 (v.2.2.1) (73) and MetaBat 2 (v.2.12.1) (74) via MetaWRAP (v.1.1.8) (75), (iii) discarded MAGs with completeness of

<80% or contamination of >5% by CheckM (v.1.0.722) (76), (iv) generated species-level genome bins (SGBs) with average nucleotide identity (ANI) set up as 95% by dRep (v.2.2.2) (77), (v) annotated the SGBs by GTDB-Tk (v.1.5.0) (78) with the Genome Taxonomy Database (GTDB) (r202) (79) as the reference database, and (vi) calculated reads per kilobase per million (RPKM) for SGBs by Coverm (https://github .com/wwood/CoverM). The fungal profiles were generated by the Kraken2 (v.2.1.1) (80) using the fungal genomes from the NCBI RefSeq with default parameters. The microbial functional profiling results were generated by an alignment-based method, HUMAnN3 (v.3.0.0), with default parameters.

**Metabolomic profiling and data processing.** Sample preparation for ultraperformance liquid chromatography-quadrupole time of flight mass spectrometry (UPLC-QTOF/MSE) analysis was performed as previously described (64). In brief, 400 $\mu$L plasma sample was added to 1,200 $\mu$L ice-methanol solution (4:1 [vol/vol]) and centrifuged at 13,000 $\times$ $g$ for 15 min at 4°C. The supernatant was collected through a 0.22-$\mu$m-pore organic filter and kept at 4°C throughout the analysis. Then, chromatographic separation and detection were achieved using an Acquity H-class UPLC system (Waters, USA) coupled with a Xevo G2-XS QTOF MS instrument (Waters, USA). Chromatographic separation was performed in the Acquity UPLC BEH C$_{18}$ column (2.1 mm by 100 mm, 1.7 $\mu$m) (Waters). To enhance metabolome coverage, ionization was carried out in both positive-ion and negative-ion modes.

The UPLC-Q-TOF MSE raw data were imported into Progenesis QI software (v.2.0) for peak matching, peak alignment, peak extraction, and normalization. Then these data were imported into R (v.4.0.5) to perform quality control and denoising. We confined the metabolites with at least 1,000 of peak intensity (and 20% prevalence in each group) and deleted the metabolites with a relative standard deviation of more than 20%. Then, the "zcomposition" package in R was used to replace the zero with pseudocount, and the "composition" package was employed to perform the centered log ratio transformation (CLR) transformation for the metabolites.

**Measuring the indirect influence of the probiotic intervention.** To explore the indirect influence, which is not considered the conventional consequence of the probiotic consumption (e.g., the abundance changes of the features in the multiomics data), we correlated the development of multiomics data and the changes of immunity indicators by different groups. In particular, we first associated individual's changes in multiomics data and the changes in advanced immune function test results using the Spearman coefficient, separately in the probiotic group and placebo groups. Then we removed the significant immunity feature (taxa, functions, or metabolites) correlations (defined by a Spearman coefficient of <0.05) that overlapped in both groups: e.g., the same positive (or negative) correlations that were identified in both groups. Finally, for the remaining features, we summed up their proportions in the probiotic group at the end of the intervention to quantify the effect size of the indirect influence.

**SNV calling.** In this study, only SNV profiles were investigated in the gut microbiome, while the insertions and deletions were not our focus. We used the *de novo* assembly-based SGBs as reference genomes to perform gut microbiome SNV calling. In particular, shotgun metagenomic sequencing reads with 100 as minimum read depth were mapped to the SGBs for SNV calling using Bowtie2 (v.2.4.4) (81) and SAMtools (v.1.10-41) (82). Candidate SNVs were identified and filtered with a minimum quality of 60 using VCFtools (v.0.1.16) (83) (https://github.com/vcftools/) and BCFtools (v.1.14-18) (84) (https://github .com/samtools/bcftools). Please refer to https://github.com/HNUmcc/Probiotics-SNV-meta for the code and more details.

We aligned the shotgun metagenomic reads from the same individuals at different time points to the same reference SGBs to identify single nucleotide changes (SNVs) by probiotic consumption. The candidate SNVs due to probiotic consumption were thought to meet the following requirements. (i) For a given SGB, a single nucleotide difference should be identified between the baseline and endpoint of a host, despite the nucleotide difference between the reference genomes. (ii) Such genetic changes can be observed in at least 30% of individuals in the study. (iii) Such genetic changes did not show up within a period that is not related to any probiotic intervention for a host. Therefore, for this study, we specifically removed SNVs that were identified as the intersections between the probiotic group and placebo group (85). For more details, please refer to the code in the GitHub repository (https://github.com/ HNUmcc/Probiotics-SNV-meta).

**Statistical analysis.** The $\alpha$ diversity values of bacterial composition, fungal composition, microbial function, and metabolites were measured by the Shannon index and calculated using the Vegan package in R 4.0.5. With the same package, Bray-Curtis dissimilarity was calculated to measure the $\beta$ diversity for the multiomics data. The differences in BC dissimilarity were determined using a nonparametric permutational multivariate analysis of variance (PERMANOVA) and visualized by UMAP using the R package umap (46). The convergence of microbial composition is measured by the within-group BC dissimilarity, and the Wilcoxon test was used to determine the significance: e.g., the Wilcoxon rank sum test was used to compare samples between the two groups at different time points, while the Wilcoxon signed-rank test was used to compare paired samples (from the same individual) between two time points in each group.

As for the differential abundance analysis, instead of simply comparing the samples from the different groups at the beginning or end of the intervention, we compared the individual development between the different groups to better identify the differential features (taxa, metabolites, and immunity indicators). First, the Wilcoxon signed-rank test was used to identify microbial species, functions, metabolites, and immunity indicators that changed during the intervention (by comparing the paired samples from the same individual at baseline and endline). Then, the significantly changed taxa, metabolites, or indicators were then compared with each other by groups: those that changed in the same way in both groups were considered the consequence of time development rather than the intervention and thus

were discarded, while the rest were considered the differentially abundant taxa, metabolites, or indicators. The differential analysis was performed using the R script in Parallel-META-3.5 (86).

**Ethics approval and consent to participate.** This study followed the Declaration of Helsinki with human subjects, and the complete procedure was approved by the Affiliated Hospital of Inner Mongolia Medical University Ethics Committee (NO.KY 2020010). Written informed consent was obtained from each participant.

**Data availability.** The data from this study have been submitted to a public, open-access repository. All sequence data from this study are available from the Genome Sequence Archive in the BIG Data Center (https://ngdc.cncb.ac.cn/gsa) under accession no. CRA009168.

## SUPPLEMENTAL MATERIAL

Supplemental material is available online only.
**SUPPLEMENTAL FILE 1**, PDF file, 0.2 MB.
**SUPPLEMENTAL FILE 2**, XLSX file, 0.1 MB.
**SUPPLEMENTAL FILE 3**, XLSX file, 0.6 MB.
**SUPPLEMENTAL FILE 4**, XLSX file, 7.3 MB.

## ACKNOWLEDGMENTS

We sincerely thank all of the volunteers in this study for their participation.

This work was supported by the National Natural Science Foundation of China (U22A20540), the Inner Mongolia Science and Technology Major Projects (2021ZD0014), the Natural Science Foundation of Inner Mongolia Autonomous Region of China (2021ZD08), and the China Agriculture Research System of MOF and MARA.

H. Zhang and W. Zhang, designed the study. Y. Zheng, C. Cao, W. Zhao, and Y. Liu collected the samples and conducted the experiments. M. Zhang, Z. Sun, and Y. Zheng analyzed the data and prepared the manuscript. Z. Sun provided suggestions and modified the manuscript. All authors approved the final version of the manuscript.

We declare no conflict of interest.

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
