## [Reviewer comments · Microbiology Spectrum]

Microbiology Spectrum

The change of the gut microbiome and immunity by *Lacticaseibacillus rhamnosus* Probio-M9

Meng Zhang, Yan Zheng, Zheng Sun, Chenxia Cao, Wei Zhao, Yangshuo Liu, Wenyi Zhang, and Heping Zhang

Corresponding Author(s): Meng Zhang, Inner Mongolia Agricultural University

Review Timeline:

Submission Date:	October 19, 2022
Editorial Decision:	November 15, 2022
Revision Received:	January 13, 2023
Accepted:	February 9, 2023

Editor: Wei-Hua Chen

Reviewer(s): Disclosure of reviewer identity is with reference to reviewer comments included in decision letter(s). The following individuals involved in review of your submission have agreed to reveal their identity: Sercan Karav (Reviewer #3); Haoyu Liu (Reviewer #4); Rustem Abuzarovich Ilyasov (Reviewer #5)

Transaction Report:

DOI: <https://doi.org/10.1128/spectrum.03609-22>

November 15, 2022

Dr. Meng Zhang
Inner Mongolia Agricultural University
Key Laboratory of Dairy Biotechnology and Engineering, Ministry of Education
Hohhot
China

Re: Spectrum03609-22 (Stabilization of the gut microbiome by *Lacticaseibacillus rhamnosus* Probio-M9)

Dear Dr. Meng Zhang:

Thank you for submitting your manuscript to Microbiology Spectrum. Your manuscript has been evaluated by three external experts. Based on their comments, I am happy to invite you to submit a revised version of your manuscript with all the reviewers' concerns properly addressed.

Link Not Available

Sincerely,

Wei-Hua Chen

Journals Department
Reviewer comments:

Reviewer #3 (Comments for the Author):

The manuscript provides an interesting data regarding the usage of probiotics and effects on human immunity and gut microbiome profile. In general, the quality of manuscript was good and the all sections were well-written. Appropriate statistical tests were applied and results were shown in detail.

Reviewer #4 (Comments for the Author):

The manuscript titled 'Stabilization of the gut microbiome by *Lactocaseibacillus rhamnosus* Probio-M9' has explored whether probiotic ingestion is beneficial for healthy individuals as a prophylaxis, by conducting a systemic analysis with large datasets. These kinds of studies should be encouraged, especially if it shows both positive and negative results. However, there are some concerns.

1. It seems that there are two arms of this study: increasing the correlation between immune parameters and microbiota changes; decreasing the genetic variation of the gut microbiome. However, the title only highlights the latter.
2. One main concern of this manuscript is its content of introduction and discussion. For the introduction, there are two paragraphs, one about the importance and the application of probiotics. One was a summary of this study (almost like the abstract). It is not sufficient to explain why it is necessary to test the effects of probiotics on host immunity and metabolism, for instance. Same problem with the discussion. One paragraph of findings summary. One paragraph about the study limitation. No specific discussion about the results. Although there are no profound changes of immunity, there are some differences. For example, the results in Fig. 1c. Perhaps there are very few studies of probiotics in healthy humans, but the microbiome, immune parameters, their meaning have had been intensively studied.
3. line 60, please revise 'increasing immunity'
4. line 193-194, it does not seem that the study involves immunity development investigation.
5. Does this manuscript contain a statistical analysis section (could not find one)? If not, it should.

Reviewer #5 (Comments for the Author):

Reviewer comments

Manuscript Spectrum03609-22 Stabilization of the gut microbiome by *Lactocaseibacillus rhamnosus* Probio-M9.

The authors found that the probiotic intervention: has a limited effect on human immunity or the global structure of the gut microbiome and metabolome; can largely influence the correlation of the development between multi-omics data and immunity, which was not able to be discovered by conventional differential abundance analysis; and inhibits the generation of adaptive SNVs in the gut microbiome instead of promoting it. The authors indicated an underestimated influence of probiotics, not on altering the microbial composition but on strengthening the association between human immunity and commensal microbes and stabilizing the genetic variations of the gut microbiome.

The data analysis methods are correct.

The English of the text is well written and well readable.

The uniqueness of the text is more than 90% by AntiPlagiarism.NET.

There are some comments and questions:

1. Why have authors investigated only SNV profiles in the gut microbiome, but not the insertions and deletions?
2. Lines 53-57 - in the sentence - For example, probiotics have been proven to be beneficial in relieving chronic diseases(4-6), improving overweight and obesity(7-10), delaying non-alcoholic fatty liver disease(11), treating gastrointestinal diseases(12-17), regulating neurological diseases(18-20), reducing subjective stress(21), improving immunity(22), and preventing antibiotic-induced *Clostridium difficile* infection(23-25). - after improving immunity(22) - add text - preventing oxidative stress and inflammation (Danilenko et al., 2021). Add to the references - Danilenko, V.N.; Devyatkin, A.V.; Marsova, M.V.; Shibilova, M.U.; Ilyasov, R.A.; Shmyrev, V.I. Common inflammatory mechanisms in COVID-19 and Parkinson's diseases: the role of microbiome, pharmabiotics and postbiotics in their prevention. *J Inflamm Res* 2021, 14, 6349-6381, doi:10.2147/JIR.S333887.
3. I did not see the hypothesis of the experiment. What did the authors intend to check as a result of the experiment? Please indicate the hypothesis of the experiment in the abstract and introduction.
4. Why do authors call them adaptive SNVs? SNVs cannot be adaptive if they are neutral. Is the inhibition of the generation of SNVs a good property of a probiotic?
5. The BioProject PRJNA844194 is not available at NCBI (<https://www.ncbi.nlm.nih.gov/>)? Please check it.
6. Please add the definition of "BMI." Check all abbreviations throughout the manuscript and add all definitions.
7. What novelty of this research? This strain *Lactocaseibacillus rhamnosus* Probio-M9 is well studied previously. Add novelty explanation to abstract and introduction.

The strain *Lactocaseibacillus rhamnosus* Probio-M9 is quite well studied in five articles in Pubmed:

Zhang J, Zhao Y, Sun Z, Sun T. *Lactocaseibacillus rhamnosus* Probio-M9 extends the lifespan of *Caenorhabditis elegans*. *Commun Biol*. 2022 Oct 27;5(1):1139. doi: 10.1038/s42003-022-04031-2. PMID: 36302976; PMCID: PMC9613993.

Zheng Y, Yu Z, Zhang W, Sun T. *Lactobacillus rhamnosus* Probio-M9 Improves the Quality of Life in Stressed Adults by Gut Microbiota. *Foods*. 2021 Oct 8;10(10):2384. doi: 10.3390/foods10102384. PMID: 34681433; PMCID: PMC8535744.

Gao G, Ma T, Zhang T, Jin H, Li Y, Kwok LY, Zhang H, Sun Z. Adjunctive Probiotic *Lactobacillus rhamnosus* Probio-M9

Administration Enhances the Effect of Anti-PD-1 Antitumor Therapy *via* Restoring Antibiotic-Disrupted Gut Microbiota. *Front Immunol*. 2021 Dec 14;12:772532. doi: 10.3389/fimmu.2021.772532. PMID: 34970262; PMCID: PMC8712698.

Xu H, Hiraishi K, Kurahara LH, Nakano-Narusawa Y, Li X, Hu Y, Matsuda Y, Zhang H, Hirano K. Inhibitory Effects of Breast Milk-Derived *Lactobacillus rhamnosus* Probio-M9 on Colitis-Associated Carcinogenesis by Restoration of the Gut Microbiota in a

Mouse Model. Nutrients. 2021 Mar 30;13(4):1143. doi: 10.3390/nu13041143. PMID: 33808480; PMCID: PMC8065529.
Liu W, Chen M, Duo L, Wang J, Guo S, Sun H, Menghe B, Zhang H. Characterization of potentially probiotic lactic acid bacteria and bifidobacteria isolated from human colostrum. J Dairy Sci. 2020 May;103(5):4013-4025. doi: 10.3168/jds.2019-17602. Epub 2020 Feb 26. PMID: 32113772. What is the novelty of this research? Add a novelty explanation to the abstract and introduction.

Please improve the manuscript according to the above comments.

Staff Comments:

Preparing Revision Guidelines

Please return the manuscript within 60 days; if you cannot complete the modification within this time period, please contact me. If you do not wish to modify the manuscript and prefer to submit it to another journal, please notify me of your decision immediately so that the manuscript may be formally withdrawn from consideration by Microbiology Spectrum.

The manuscript titled ‘Stabilization of the gut microbiome by Lacticaseibacillus rhamnosus Probio-M9’ has explored whether probiotic ingestion is beneficial for healthy individuals as a prophylaxis, by conducting a systemic analysis with large datasets. These kind of studies should be encouraged, especially it shows both positive and negative results. However, there are some concerns.

1. It seems that there are two arms of this study: increasing the correlation between immune parameters and microbiota changes; decreasing the genetic variation of the gut microbiome. However, the title only highlights the latter.

2. One main concern of this manuscript is its content of introduction and discussion. For introduction, there are two paragraphs, one about the importance and the application of the probiotics. One was a summary of this study (almost like the abstract). It is not sufficient to explain why it is necessary to test the effects of probiotics on host immunity and metabolism, for instance. Same problem with the discussion. One paragraph of findings summary. One paragraph about the study limitation. No specific discussion about results. Although there are no profound changes of immunity, there are some difference. For example, the results in Fig. 1c. Perhaps there are very few studies of probiotics in healthy humans, but the microbiome, immune parameters, their meaning have had been intensively studied.

3. line 60, please revise ‘increasing immunity’

4. line193-194, it does not seem that the study involves immunity development investigation.

5. Does this manuscript contain a statistical analysis section (could not find one)? If not, it should.

Reviewer comments

Manuscript Spectrum03609-22 Stabilization of the gut microbiome by *Lacticaseibacillus rhamnosus* Probio-M9.

The authors found that the probiotic intervention: has a limited effect on human immunity or the global structure of the gut microbiome and metabolome; can largely influence the correlation of the development between multi-omics data and immunity, which was not able to be discovered by conventional differential abundance analysis; and inhibits the generation of adaptive SNVs in the gut microbiome instead of promoting it. The authors indicated an underestimated influence of probiotics, not on altering the microbial composition but on strengthening the association between human immunity and commensal microbes and stabilizing the genetic variations of the gut microbiome.

The data analysis methods are correct.

The English of the text is well written and well readable.

The uniqueness of the text is more than 90% by AntiPlagiarism.NET.

There are some comments and questions:

1. Why have authors investigated only SNV profiles in the gut microbiome, but not the insertions and deletions?
2. Lines 53-57 - in the sentence - For example, probiotics have been proven to be beneficial in relieving chronic diseases(4-6), improving overweight and obesity(7-10), delaying non-alcoholic fatty liver disease(11), treating gastrointestinal diseases(12-17), regulating neurological diseases(18-20), reducing subjective stress(21), improving immunity(22), and preventing antibiotic-induced *Clostridium difficile* infection(23-25). - after improving immunity(22) - add text - preventing oxidative stress and inflammation (Danilenko et al., 2021). Add to the references - Danilenko, V.N.; Devyatkin, A.V.; Marsova, M.V.; Shibilova, M.U.; Ilyasov, R.A.; Shmyrev, V.I. Common inflammatory mechanisms in COVID-19 and Parkinson's diseases: the role of microbiome, pharmabiotics and postbiotics in their prevention. *J Inflamm Res* 2021, 14, 6349–6381, doi:10.2147/JIR.S333887.
3. I did not see the hypothesis of the experiment. What did the authors intend to check as a result of the experiment? Please indicate the hypothesis of the experiment in the abstract and introduction.
4. Why do authors call them adaptive SNVs? SNVs cannot be adaptive if they are neutral. Is the inhibition of the generation of SNVs a good property of a probiotic?
5. The BioProject PRJNA844194 is not available at NCBI (<https://www.ncbi.nlm.nih.gov>)? Please check it.
6. Please add the definition of "BMI." Check all abbreviations throughout the manuscript and add all definitions.
7. What novelty of this research? This strain *Lacticaseibacillus rhamnosus* Probio-M9 is well studied previously. Add novelty explanation to abstract and introduction.
The strain *Lacticaseibacillus rhamnosus* Probio-M9 is quite well studied in five articles in Pubmed:

Zhang J, Zhao Y, Sun Z, Sun T. *Lacticaseibacillus rhamnosus* Probio-M9 extends the lifespan of *Caenorhabditis elegans*. *Commun Biol*. 2022 Oct 27;5(1):1139. doi: 10.1038/s42003-022-04031-2. PMID: 36302976; PMCID: PMC9613993.

Zheng Y, Yu Z, Zhang W, Sun T. *Lactobacillus rhamnosus* Probio-M9 Improves the Quality of Life in Stressed Adults by Gut Microbiota. *Foods*. 2021 Oct 8;10(10):2384. doi: 10.3390/foods10102384. PMID: 34681433; PMCID: PMC8535744.

Gao G, Ma T, Zhang T, Jin H, Li Y, Kwok LY, Zhang H, Sun Z. Adjunctive Probiotic *Lactobacillus rhamnosus* Probio-M9 Administration Enhances the Effect of Anti-PD-1 Antitumor Therapy via Restoring Antibiotic-Disrupted Gut Microbiota. *Front Immunol.* 2021 Dec 14;12:772532. doi: 10.3389/fimmu.2021.772532. PMID: 34970262; PMCID: PMC8712698.

Xu H, Hiraishi K, Kurahara LH, Nakano-Narusawa Y, Li X, Hu Y, Matsuda Y, Zhang H, Hirano K. Inhibitory Effects of Breast Milk-Derived *Lactobacillus rhamnosus* Probio-M9 on Colitis-Associated Carcinogenesis by Restoration of the Gut Microbiota in a Mouse Model. *Nutrients.* 2021 Mar 30;13(4):1143. doi: 10.3390/nu13041143. PMID: 33808480; PMCID: PMC8065529.

Liu W, Chen M, Duo L, Wang J, Guo S, Sun H, Menghe B, Zhang H. Characterization of potentially probiotic lactic acid bacteria and bifidobacteria isolated from human colostrum. *J Dairy Sci.* 2020 May;103(5):4013-4025. doi: 10.3168/jds.2019-17602. Epub 2020 Feb 26. PMID: 32113772. What is the novelty of this research? Add a novelty explanation to the abstract and introduction.

Please improve the manuscript according to the above comments.

Wei-Hua Chen, PhD, Editor,

Microbiology Spectrum

January 11, 2023

Dear Dr. Chen,

Thank you and the reviewers for commenting on our manuscript and giving us the opportunity to revise. Here we have modified the manuscript and addressed all the comments as detailed below. All revisions in the manuscript are highlighted, and the file is submitted as “Marked Up Manuscript - For Review Only”.

Responses to Reviewer #3

Reviewer comments:

Reviewer #3 (Comments for the Author):

The manuscript provides an interesting data regarding the usage of probiotics and effects on human immunity and gut microbiome profile. In general, the quality of manuscript was good and the all sections were well-written. Appropriate statistical tests were applied and results were shown in detail.

Response 1: We thank Reviewer #3 very much for reviewing our manuscript and her/his very positive assessment of our work.

Responses to Reviewer #4

Reviewer #4 (Comments for the Author):

*The manuscript titled 'Stabilization of the gut microbiome by *Lactocaseibacillus rhamnosus* Probio-M9' has explored whether probiotic ingestion is beneficial for healthy individuals as a prophylaxis, by conducting a systemic analysis with large datasets. These kinds of studies should be encouraged, especially if it shows both positive and negative results. However, there are some concerns.*

Response 2: We thank Reviewer #4 very much for reviewing our manuscript and her/his very positive assessment of the general interest of our work. We have reorganized many parts of the manuscript to improve the scientific quality, and we have addressed each of the Reviewer’s comments in order.

1. It seems that there are two arms of this study: increasing the correlation between immune parameters and microbiota changes; decreasing the genetic variation of the gut microbiome. However, the title only highlights the latter.

Response 3: We thank Reviewer #4 for pointing out our issues. We have reorganized the title: “The change of the gut microbiome and immunity by *Lactocaseibacillus rhamnosus* Probio-M9”, which we think can cover the two arms in our study in a simple but brief manner.

2. One main concern of this manuscript is its content of introduction and discussion. For the introduction, there are two paragraphs, one about the importance and the application of probiotics. One was a summary of this study (almost like the abstract). It is not sufficient to explain why it is necessary to test the effects of probiotics on host immunity and metabolism, for instance. Same problem with the discussion. One paragraph of findings summary. One paragraph about the study limitation. No specific discussion about the results. Although there are no profound changes of immunity, there are some differences. For example, the results in Fig. 1c. Perhaps there are very few studies of probiotics in healthy humans, but the microbiome, immune parameters, their meaning have had been intensively studied.

Response 4: We thank Reviewer #4 for pointing out the issues. We have modified the introduction (the first paragraph below) and discussion sections (the second paragraph below) as below:

“Probiotics, defined as live microorganisms which when administered in adequate amounts confer a health benefit on the host (1, 2), are one of the most clinically feasible approaches for regulating the gut microbiome and metabolome(3), and have a great potential to be therapeutic targets in many human diseases. For example, *evidence from previous studies have suggested that modulation of the intestinal microbiota and microbial metabolites by consuming probiotic is beneficial in relieving chronic diseases(4-6), improving overweight and obesity(7-10), delaying non-alcoholic fatty liver disease(11), treating gastrointestinal diseases(12-17), regulating neurological diseases(18-20), reducing subjective stress(21), improving immunity(22), preventing oxidative stress and inflammation(23), and preventing antibiotic-induced Clostridium difficile infection(24-26).* However, an active debate has been going on for years about whether the consumption of probiotic is beneficial for nonpatients in life quality improvement and disease prevention(27-29). Although previous studies supported a beneficial effect of probiotic in enhance immunity against the common cold, which can reduce the incidence(30), duration(31), and symptoms(32) of the common cold(28, 33, 34), some reported that the effects of probiotic on the immune system and gastrointestinal symptoms in nonpatients are limited(29, 35). *One approach by which probiotics impact the host immune system is through the microbial metabolism that arises from intestinal microbiota catabolism(36-39), e.g., functional metagenomic studies have identified the associations between host pro-inflammatory cytokines and microbial tryptophan and palmitoleic acid metabolic pathways(37, 38, 40-42).* Moreover, previous studies have illustrated that the gut microbiota and metabolite activate the host immune system, thereby increasing the expression of endocrine peptides and promoting host metabolic homeostasis(37, 43, 44). In addition, the characterization of stable and changeable genetic components in the gut microbiome is crucial for further understanding the role of the gut microbiome in human health and phenotypic changes (45), which was rarely been investigated in probiotic studies. Therefore, a comprehensive evaluation of *the role of gut microbiome, the change of microbial metabolites, and variations of their genetic structure in the beneficial effect of probiotic consumption is still urgently warranted due to the lack of systematic analysis of objective immunity indicators and time-series multi-omics data in previous studies.*”

...

“...Therefore, there is an urgent need in understanding the benefits of probiotics on nonpatients through immunity indicators, the gut microbiome, and microbial metabolism using time-series data in

a quantitative and systematic way.

The functional immune system is a central determinant of survival of the host organismal health from environmental pathogens, viruses, or chemicals(53, 54). Although there is just slightly altered overall human immunity after the probiotic intervention compared to the placebo, six cytokines such as Eotaxin, IL-1 α , IL-8, MCP-1, RANTES TNF- β , and one lymphocyte CD3+CD8+ T cells showed significant improvements within the normal range, which is in line with previous findings that the T lymphocytes and cytokines interleukin are activated after probiotic consumption(28, 31, 55, 56). Complex ecological communities are generally thought to be more stable and resilient, and gut microbial diversity is often used as a proxy for human health (58-60). Even though six weeks of Lactacaseibacillus rhamnosus Probio-M9 consumption did not dramatically affect the overall gut microbiome and metabolites, it reduced the population heterogeneity of individuals in view of the multi-omics perspective. To better understand such impacts on the host, we innovatively evaluated the probiotic's effects by categorizing them into direct effects and indirect effects. As expected, we found a significantly underestimated indirect effect of probiotics on the other intestinal microorganisms, metabolites, and immune indicators, underling the intermediary effect of Probio-M9 that regulates intestinal microbes and microbial metabolites and eventually affect the host immune system.”

3. line 60, please revise 'increasing immunity'

Response 5: We thank Reviewer #4 for this comment. We have revise ‘increasing immunity’ into ‘enhanced immunity’

4. line193-194, it does not seem that the study involves immunity development investigation.

Response 6: We thank Reviewer #4 for this comment and sorry for misleading. We have changed the subtitle to: “The underestimated influence of probiotic intake on the association between multi-omics data and immunity” to avoid the confusion of “immunity development” and “change of immunity related indexes”.

5. Does this manuscript contain a statistical analysis section (could not find one)? If not, it should.

Response 7: We thank Reviewer #4 for this comment. We didn’t provide an independent “Statistical analysis” section. Rather, we described those details in each section, e.g., the alpha diversity and beta diversity analysis methods in the “Alpha and beta diversity analysis” section; the differentially abundant species taxa identification in the “Differential abundance analysis” section. Now, we have added a “Statistical analysis” section in our revised manuscript, it reads:

“The alpha diversity of bacterial composition, fungal composition, microbial function, and metabolites were measured by the Shannon index and calculated using the “Vegan” package in R 4.0.5. With the same package, Bray-Curtis dissimilarity was calculated to measure the beta diversity for the multi-omics data. The differences in BC dissimilarity were determined using a non-parametric multivariate analysis of variance (Permanova test) and visualized by the UMAP using R package “umap”(46). The convergence of microbial composition is measured by the within group BC

dissimilarity, and the Wilcoxon test was used to determine the significance, e.g., the Wilcoxon Rank-sum test was used to compare samples between the two groups at different timepoints while the Wilcoxon Signed-rank test was used to compare paired samples (from the same individual) between two timepoints in each group.

As for the differential abundance analysis, instead of simply comparing the samples from the different groups at the beginning or end of the intervention, we compared the individual development between the different groups to better identify the differentially features (taxa, metabolites, and immunity indicators). Firstly, the Wilcoxon Sign-rank test was first used to identify microbial species, functions, metabolites, and immunity indicators that changed during the intervention (by comparing the paired samples from the same individual at baseline and endline). Then, the significantly changed taxa, metabolites, or indicators were then compared with each other by groups: those who changed in the same way in both groups were considered as the consequence of time development rather than the intervention and thus were discarded while the rest were considered as the differentially abundant taxa, metabolites, or indicators. The differential analysis was performed using the R script in Parallel-META-3.5(86).”

Reviewer #5 (Comments for the Author):

The authors found that the probiotic intervention: has a limited effect on human immunity or the global structure of the gut microbiome and metabolome; can largely influence the correlation of the development between multi-omics data and immunity, which was not able to be discovered by conventional differential abundance analysis; and inhibits the generation of adaptive SNVs in the gut microbiome instead of promoting it. The authors indicated an underestimated influence of probiotics, not on altering the microbial composition but on strengthening the association between human immunity and commensal microbes and stabilizing the genetic variations of the gut microbiome.

The data analysis methods are correct.

The English of the text is well written and well readable.

The uniqueness of the text is more than 90% by AntiPlagiarism.NET.

Response 8: We thank Reviewer #5 very much for reviewing our manuscript and her/his very positive assessment on the novelty and relevance of our work. We next address each of the comments in order.

There are some comments and questions:

1. Why have authors investigated only SNV profiles in the gut microbiome, but not the insertions and deletions?

Response 9: We thank Reviewer #5 for this insightful comment. We only included the SNVs in our analysis simply because: (i) SNV is a powerful measurement for adaptive evolution. For example, many studies reported that a single SNV in the microbial genome could significantly alter the pathogenic behavior of gut bacteria and affect host health(1-3). Moreover, Chen et al(4). identified that specific SNVs on the genome of *Bacteroides coprocola* were correlated with T2D; Zou et al(2). reported that BlcE84-encoding bacteria with a distinctive SNV on the genome caused the destruction of the worm and mouse epithelial barrier and immune activation. (ii) Insertions and deletions largely

rely on the sequencing techniques and bioinformatics tools while these will influence the sensitivity and specificity of indel detection and may lead to false positives(5-9). Previous studies have shown that the reversible dye terminator approach and short-length reads of Illumina sequencing platforms will influence the indel error rate(6, 7). In addition, the lack of consensus standards (for detection and annotation) is still the primary limitation for indel detection(7, 8).

2. Lines 53-57 - in the sentence - For example, probiotics have been proven to be beneficial in relieving chronic diseases(4-6), improving overweight and obesity(7-10), delaying non-alcoholic fatty liver disease(11), treating gastrointestinal diseases(12-17), regulating neurological diseases(18-20), reducing subjective stress(21), improving immunity(22), and preventing antibiotic-induced Clostridium difficile infection(23-25). - after improving immunity(22) - add text - preventing oxidative stress and inflammation (Danilenko et al., 2021). Add to the references - Danilenko, V.N.; Devyatkin, A.V.; Marsova, M.V.; Shibilova, M.U.; Ilyasov, R.A.; Shmyrev, V.I. Common inflammatory mechanisms in COVID-19 and Parkinson's diseases: the role of microbiome, pharmabiotics and postbiotics in their prevention. J Inflamm Res 2021, 14, 6349-6381, doi:10.2147/JIR.S333887.

Response 10: We thank Reviewer #5 for this critical comment. We have added the content as requested.

3. I did not see the hypothesis of the experiment. What did the authors intend to check as a result of the experiment? Please indicate the hypothesis of the experiment in the abstract and introduction.

Response 11: We thank Reviewer #5 for this constructive comment. Our central hypothesis is that the consumption of probiotics (such as *Lactobacillus rhamnosus M9*) may influence immunity and its association with the gut microbiome, though with limited change for the latter. We have indicated the hypothesis in the abstract (the first paragraph below) and introduction (the second paragraph below).

*“In this study, we recruited 100 adults from a college in China and performed a random case-control study by using a probiotic (*Lacticaseibacillus rhamnosus* Probio-M9) as an intervention for six weeks, aiming to make a comprehensively evaluation and understanding of the beneficial effect by the Probio-M9 consumption.”*

...

*“To address the above questions, we recruited 100 adult volunteers in China, and randomly assigned them into two groups. Then, *Lacticaseibacillus rhamnosus* Probio-M9 (5.0×10^{10} CFU per day) was administrated as the probiotic intervention in one of the groups for six weeks. We sought to explore how the *Lactobacillus rhamnosus* Probio-M9 would impact the gut microbiome, gut metabolome, host immune response, and its variants in the genomic structure after consumption in humans. So, we measured advanced immune function tests, metagenomic sequencing, and metabolomic profiling were performed using the blood and stool samples collected at the beginning (baseline) and end of the trial (endline).”*

4. Why do authors call them adaptive SNVs? SNVs cannot be adaptive if they are neutral. Is the inhibition of the generation of SNVs a good property of a probiotic?

Response 12: We thank Reviewer #5 for pointing out this issue. We are sorry about the misleading. Indeed, adaptive evolution can only be detected by the independent recurrence of similar mutations in genes under selection (parallel evolution or convergent evolution) or by an increase in mutational frequency that is inconsistent with neutral drift. To avoid the misleading, we have replaced “adaptive SNVs” with “SNVs” throughout the manuscript.

5. The BioProject PRJNA844194 is not available at NCBI (<https://www.ncbi.nlm.nih.gov>)? Please check it.

Response 13: We thank Reviewer #5 for bringing up the issue. To fix this, we have resubmitted the raw sequencing data into the Genome Sequence Archive in BIG Data Center (<https://ngdc.cncb.ac.cn/gsa>) under the accession number CRA009168 (<https://ngdc.cncb.ac.cn/gsa/s/FX1wF73h>), and revised the data availability section accordingly.

6. Please add the definition of "BMI." Check all abbreviations throughout the manuscript and add all definitions.

Response 14: We thank Reviewer #5 for this comment. We have added the definition of the BMI (Body mass index) in our revised manuscript, and we have checked all abbreviations throughout the manuscript and supplemented the definitions.

7. What novelty of this research? This strain Lactocaseibacillus rhamnosus Probio-M9 is well studied previously. Add novelty explanation to abstract and introduction.

The strain Lactocaseibacillus rhamnosus Probio-M9 is quite well studied in five articles in Pubmed: Zhang J, Zhao Y, Sun Z, Sun T. Lactocaseibacillus rhamnosus Probio-M9 extends the lifespan of Caenorhabditis elegans. Commun Biol. 2022 Oct 27;5(1):1139. doi: 10.1038/s42003-022-04031-2. PMID: 36302976; PMCID: PMC9613993.

Zheng Y, Yu Z, Zhang W, Sun T. Lactobacillus rhamnosus Probio-M9 Improves the Quality of Life in Stressed Adults by Gut Microbiota. Foods. 2021 Oct 8;10(10):2384. doi: 10.3390/foods10102384. PMID: 34681433; PMCID: PMC8535744.

Gao G, Ma T, Zhang T, Jin H, Li Y, Kwok LY, Zhang H, Sun Z. Adjunctive Probiotic Lactobacillus rhamnosus Probio-M9 Administration Enhances the Effect of Anti-PD-1 Antitumor Therapy via Restoring Antibiotic-Disrupted Gut Microbiota. Front Immunol. 2021 Dec 14;12:772532. doi: 10.3389/fimmu.2021.772532. PMID: 34970262; PMCID: PMC8712698.

Xu H, Hiraishi K, Kurahara LH, Nakano-Narusawa Y, Li X, Hu Y, Matsuda Y, Zhang H, Hirano K. Inhibitory Effects of Breast Milk-Derived Lactobacillus rhamnosus Probio-M9 on Colitis-Associated Carcinogenesis by Restoration of the Gut Microbiota in a Mouse Model. Nutrients. 2021 Mar 30;13(4):1143. doi: 10.3390/nu13041143. PMID: 33808480; PMCID: PMC8065529.

Liu W, Chen M, Duo L, Wang J, Guo S, Sun H, Menghe B, Zhang H. Characterization of potentially probiotic lactic acid bacteria and bifidobacteria isolated from human colostrum. J Dairy Sci. 2020 May;103(5):4013-4025. doi: 10.3168/jds.2019-17602. Epub 2020 Feb 26. PMID: 32113772. What is the novelty of this research? Add a novelty explanation to the abstract and introduction.

Response 15: We thank Reviewer #5 for this constructive comment. Liu's study demonstrated the isolation of the *Lacticaseibacillus rhamnosus* Probio-M9 and its biochemical characteristics. Zhang, Gao, and Xu's study illustrated the beneficial effect of Probio-M9 to the host using animal models, thus it is still not clear whether the beneficial effect can be acquired by humans. Although Zheng's study involved humans, the sample size is so limited and only metagenomic data were used. Additionally, Zheng and his/her colleagues only investigated the performance of Probio-M9 on improving the life quality of stressed adults, their research didn't reveal any comprehensive understanding of the mechanism, e.g., how the Probio-M9 influences the gut microbiome and metabolome, how the host immune response to the probiotic, and its variants in the genomic structure after consumption. We have added the statements of novelty into the abstract, introduction (please see Response #11), and discussion as advised.

*“To our knowledge, the present research is the first to report the underestimated alteration of association between immunity and the gut microbiome/metabolome, as well as the suppressed SNVs by probiotic consumption in nonpatients. Despite many studies that have been conducted on *Lacticaseibacillus rhamnosus* Probio-M9 (such as its isolation process, biochemical characteristics (60), and mouse(61-63) and small-scale human experiments(64) examining its beneficial effects on hosts), the comprehensive evaluation of Probio-M9 using multi-omics data collected from different timepoints, as presented in this paper, greatly compensates for the gaps in probiotics research and significantly increases our understanding of the mechanisms behind the probiotic beneficial effects.”*

Reference cited in this response letter:

1. Sokurenko EV, Chesnokova V, Dykhuizen DE, Ofek I, Wu XR, Krogfelt KA, Struve C, Schembri MA, Hasty DL. 1998. Pathogenic adaptation of *Escherichia coli* by natural variation of the FimH adhesin. *Proc Natl Acad Sci U S A* 95:8922-6.
2. Zou D, Pei J, Lan J, Sang H, Chen H, Yuan H, Wu D, Zhang Y, Wang Y, Wang D, Zou Y, Chen D, Ren J, Gao X, Lin Z. 2020. A SNP of bacterial *blc* disturbs gut lysophospholipid homeostasis and induces inflammation through epithelial barrier disruption. *EBioMedicine* 52:102652.
3. Zhu Q, Hou Q, Huang S, Ou Q, Huo D, Vázquez-Baeza Y, Cen C, Cantu V, Estaki M, Chang H, Belda-Ferre P, Kim HC, Chen K, Knight R, Zhang J. 2021. Compositional and genetic alterations in Graves' disease gut microbiome reveal specific diagnostic biomarkers. *ISME J* 15:3399-3411.
4. Chen YW, Li ZC, Hu SF, Zhang J, Wu JQ, Shao NS, Bo XC, Ni M, Ying XM. 2017. Gut metagenomes of type 2 diabetic patients have characteristic single-nucleotide polymorphism distribution in *Bacteroides coprocola*. *Microbiome* 5.
5. Quail MA, Smith M, Coupland P, Otto TD, Harris SR, Connor TR, Bertoni A, Swerdlow HP, Gu Y. 2012. A tale of three next generation sequencing platforms: comparison of Ion Torrent, Pacific Biosciences and Illumina MiSeq sequencers. *BMC genomics* 13:1-13.
6. Sehn JK. 2015. Insertions and deletions (indels), p 129-150, *Clinical genomics*. Elsevier.
7. Fang H, Bergmann EA, Arora K, Vacic V, Zody MC, Iossifov I, O'Rawe JA, Wu Y, Jimenez Barron LT, Rosenbaum J. 2016. Indel variant analysis of short-read sequencing data with Scalpel. *Nature protocols* 11:2529-2548.

8. Bennett EP, Petersen BL, Johansen IE, Niu Y, Yang Z, Chamberlain CA, Met Ö, Wandall HH, Frödin M. 2020. INDEL detection, the 'Achilles heel' of precise genome editing: a survey of methods for accurate profiling of gene editing induced indels. *Nucleic Acids Research* 48:11958-11981.
9. Foox J, Tighe SW, Nicolet CM, Zook JM, Byrska-Bishop M, Clarke WE, Khayat MM, Mahmoud M, Laaguiby PK, Herbert ZT. 2021. Performance assessment of DNA sequencing platforms in the ABRF Next-Generation Sequencing Study. *Nature biotechnology* 39:1129-1140.
10. Chen LM, Wang DM, Garmaeva S, Kurilshikov A, Vila AV, Gacesa R, Sinha T, Segal E, Weersma RK, Wijmenga C, Zhernakova A, Fu JY. 2021. The long-term genetic stability and individual specificity of the human gut microbiome. *Cell* 184:2302-+.

Again, thanks to the anonymous reviewers for the constructive comments that have improved our manuscript. If you have any additional suggestions, please do not hesitate to contact me via phone call or email.

Heping Zhang, Ph.D.

Professor of Chang Jiang Program

Professor, Key Laboratory of Dairy Biotechnology and Engineering, Ministry of Education, Inner Mongolia Agricultural University, China

Inner Mongolia Agricultural University, 306 Zhaowuda Rd, Huhhot, Inner Mongolia, China

Email: hepingdd@vip.sina.com

Tel: (86) 0471- 4319940, Fax: (86) 0471-4305357

February 9, 2023

Dr. Meng Zhang
Inner Mongolia Agricultural University
Key Laboratory of Dairy Biotechnology and Engineering, Ministry of Education
Hohhot
China

Re: Spectrum03609-22R1 (The change of the gut microbiome and immunity by *Lacticaseibacillus rhamnosus* Probio-M9)

Dear Dr. Meng Zhang:

Thank you for submitting your manuscript to our journal. Your revision has been evaluated by two external experts. It is their consensus that the manuscript can be accepted. I am glad to inform you that your manuscript has been accepted, and I am forwarding it to the ASM Journals Department for publication. You will be notified when your proofs are ready to be viewed.

Sincerely,

Wei-Hua Chen
Editor, Microbiology Spectrum

Journals Department
Reviewer comments Spectrum03609-22R1

Reviewer Recommendation and Comments for the Manuscript Spectrum03609-22R1
The change of the gut microbiome and immunity by *Lacticaseibacillus*
rhamnosus Probio-M9

I carefully read the revised manuscript. All requirements of the reviewers are met.

The manuscript has been improved after revision significantly.

There are no additional comments for authors.

Great job.

I recommend the manuscript for acceptance.

This manuscript can be published in the journal Microbiology Spectrum.